# Research on the Application of Palm Mat Geotextiles for Sand Fixation in the Hobq Desert

**Shuai Zhong** [1,*]**, Zhiwen Han** [1]**, Aimin Li** [1,2] **and Heqiang Du** [1]

[1]   Key Laboratory of Desert and Desertification, Northwest Institute of Eco-Environment and Resources, CAS, Lanzhou 730000, China; hzwen@lzb.ac.cn (Z.H.); aiminliok@126.com (A.L.); dilikexue119@163.com (H.D.)

[2]   Resources and Environment Department, Heze University, Heze 274015, China

\*   Correspondence: zhongshuai20000@163.com; Tel.: +86-0931-4967542

**Abstract:** Traditional sand fixation measures have many limitations. For example, engineering sand fixation measures using barriers cannot completely stabilize sand dunes. Biological sand fixation measures utilizing planted vegetation are very difficult to build during the early stages of desertification control. Chemical sand fixation measures spray chemically bonded materials on sand dunes to form a consolidated layer to prevent sand flow, but the cost of the materials used is relatively high. Therefore, new sand fixation technologies and methods urgently need to be developed. This study demonstrates a new sand fixation method, which uses palm mat geotextiles to stabilize sand dunes and to plant grass. We investigated the physical properties of these palm mat geotextiles in the laboratory and observed vegetation growth in the Hobq Desert. The results showed the following: (1) Palm mat geotextiles are lighter and tougher than common straw mat geotextiles. The average weight, thickness, and tensile strength of palm mat geotextiles are 2023 g/m$^2$, 20.14 mm, and 842–860 kPa, respectively. After a year of field observations, the tensile strength decreased by only 2%. (2) Palm mat geotextiles have excellent water retention capacity and scouring resistance; the maximum water content reached 227%, and the substrate lost 2.9% after laboratory simulation of heavy rainfall for three hours with a rainfall intensity of 30 mm h$^{-1}$. (3) Palm mat geotextiles significantly decreased the soil temperature and increased moisture in summer. The results showed that the palm mat geotextiles had the largest influence on soil temperature in the upper 5 cm of soil and the largest influence on soil moisture in the upper 10 cm of soil. (4) The field experimental results showed that, by the end of the experiment, the vegetation coverage and the biomass of the palm mat geotextiles with dimensions of 2 × 2 m were 3.9 times and 4.1 times that of the control group and 1.7 times and 1.8 times that of the group of high-banded *Salix psammophila* sand barriers, which are widely used in the Hobq Desert at present. Palm mat geotextiles are a promising material for sand fixation in the Hobq Desert.

**Keywords:** growth promotion; palm mat geotextiles; physical property; sand fixation

---

## 1. Introduction

According to the definition given by the United Nations Convention to Combat Desertification, desertification refers to land degradation in arid, semiarid, and dry subhumid areas resulting from various factors including climatic variations and human activities [1]. Desertification is a very serious environmental and socioeconomic problem facing the world. China is one of the most seriously desertified countries in the world, with a desertified area covering $1.72 \times 10^6$ km$^2$. This area involves 30 provinces (regions) and accounts for approximately 17.93% of the territorial area as of the end of 2014 [2]. There are several major types of desertification, including sandy desertification, soil and water erosion, and salinization [3]. Sandy desertification, or aeolian desertification [4], is the most severe issue among all types of desertification present in Northern China. In this type of desertification,

the fragility of ecosystems is predetermined by inherently harsh physical conditions, such as sparse vegetation, a continental climate, sandy soils, and water deficiency [5–7]. Sand fixation is one of the major measures used for sandy desertification control.

Generally, there are three kinds of sand fixation measures: engineering sand fixation measures, biological sand fixation measures, and chemical sand fixation measures. Engineering sand-fixation measures involves, for example, branches [8], plant straw [9], and gravel [10,11], which are used to set up barriers to fix the sand dunes; materials more recently used include nenolen nets [12], HDPE nets [13], and environmental plant fiber nets. Biological sand fixation measures stabilize the sand dunes by planting vegetation [14,15], which is an economical, persistent, and effective method, but vegetation is very difficult to cultivate in the early stages of desertification control [16–18]. Chemical sand fixation measures consist of spraying chemically bonded materials on the sand dunes to form a consolidated layer to prevent sand movement [19], but the cost of materials is relatively high [20]. For example, a typical representative of engineering measures is the high-banded *Salix psammophila* sand barriers, which are widely used in the Hobq Desert and cost approximately 3.5 CNY per square meter. Chemical sand fixation measures using ester copolymer cost about 4.5 CNY per square meter. Although some progress has been made recently to address sand fixation around the world, there are still many problems to be solved, such as the lack of new technologies and new materials.

Geotextile sand fixation measures combine engineering sand fixation measures and biological sand fixation measures by using plant fabrics as the carrier, which contains seeds (of trees, shrubs, and grass) and nutrient soil and is fixed by a fabric net [21–23]. Geotextiles are widely used in landscaping, road and railway construction, reservoir and river embankments, and other slope greening protections [24–28].

Palm fiber is strong, wear-resistant, breathable, light, corrosion-resistant, and elastic, and is a new type of natural fiber now starting to be used in many fields. Palm mat geotextiles are used for rain splash erosion control [29]. They are also are used for conservation of loamy sand soils in East Shropshire, UK [30]. Palm and simulated geotextiles are used to reduce runoff and interrill erosion on medium and steep slopes [31].

However, there are no reports regarding the use of palm mat geotextiles for sand fixation measures in the desert. To address this, we experimented with palm mat geotextiles for the stabilization of sand dunes in the Hobq Desert. The Hobq Desert covers approximately $1.45 \times 10^4$ km$^2$ and is the seventh largest desert in China; it is located in the northern portion of the Ordos Plateau. This desert is an important sandstorm source threatening the Beijing, Tianjin, and Hebei urban regions, which are approximately 800 km from the Hobq Desert. In this respect, it is of great urgency to control the sandy desertification of the Hobq Desert.

The objective of this study is to evaluate the effect of sand fixation using palm mat geotextiles compared with traditional sand fixation measures and to find new, inexpensive, environmentally conscious and more suitable sand fixation materials for sandy desertification control in the Hobq Desert.

## 2. Materials and Methods

### 2.1. Materials

Palm mat geotextiles use palm fiber as the carrier and contain a variety of materials, including seeds, peat soil, nonwoven fabric, and a fixed net, and these were designed by our research team and produced by KLD International Environmental Vegetation (Beijing) Co., Ltd. These palm mat geotextiles included two fixed net layers, a palm fiber layer, a seed and nutrient soil layer, and a nonwoven fabric layer, and these layers were sutured via large machinery at one time (Figure 1). The geotextiles were made of palm fibers, polypropylene, and nonwoven fabric materials. The palm fiber was 15–25 cm long. The fixed net was made of polypropylene and had dimensions of $5 \times 5$ mm. The weight of the nonwoven fabric was 100 g/m$^2$. The sewing thread was made of polypropylene where d $\geq$ 0.3 mm (Table 1). Seeds were selected based on local plant species, including *Agriophyllum*

*squarrosum* (L.) Moq., *Artemisia desertorum* Spreng. Syst. Veg., *Hedysarum mongolicum* Turcz., and *Hedysarum scoparium* Fisch. et Mey. (Table 2). The distribution of the seeds was 0.53 g/m$^2$, which was based on the distribution of aerial seeding afforestation in the Hobq Desert. The mass proportion of the four seeds was 1:1:2:2. The peat soil layer was set up to promote plant growth. The weight of the peat soil was 100 g/m$^2$ (Table 3).

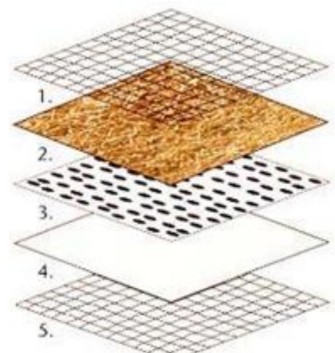

**Figure 1.** Structure of the palm mat geotextiles. The layers of the palm mat geotextiles include two fixed net layers, a palm fiber layer, a seed and nutrient soil layer, and a nonwoven fabric layer, and these were sutured with large machinery at one time. 1. Fixed net; 2. Palm fiber layer; 3. Seed and nutrient soil layer; 4. Nonwoven fabric layer; 5. Fixed net.

**Table 1.** Composition and specifications of materials used in palm mat geotextiles.

| Types | Materials | Specifications |
|---|---|---|
| Raw material | Palm fiber | 15–25 cm |
| Cushion | Nonwoven fabric | 100 g/m$^2$ |
| Fixed net | Polypropylene (PP) | 5 × 5 mm |
| Sewing thread | Polypropylene (PP) | d ≥ 0.3 mm |

**Table 2.** The germinability (at 25 °C), seed weight and purity of seeds sown in nutrient soil layer.

| Species | Germination Ration (%) | 1000 Seed Weight (g) | Purity (%) |
|---|---|---|---|
| *Agriophyllum squarrosum* (L.) Moq. | 79.77 | 0.92 | 92.75 |
| *Artemisia desertorum* Spreng. Syst. Veg. | 81.35 | 0.86 | 89.73 |
| *Hedysarum mongolicum* Turcz. | 67.67 | 27.44 | 91.62 |
| *Hedysarum scoparium* Fisch. et Mey. | 72.33 | 16.35 | 93.47 |

Seed was sown at 0.53 g/m$^2$ at a ration of 1:1:2:2 *A. squarrosum*: *A. desertorum*: *H. mongolicum*: *H. scoparium*.

**Table 3.** Properties of the peat soil layer (mg/kg).

| pH | $CO_3^{2-}$ | $SO_4^{2-}$ | $CL^-$ | $HCO_3^-$ | $Ca^{2+}$ | N | P | K | Organic Matter |
|---|---|---|---|---|---|---|---|---|---|
| 7.85 | 6.63 | 0.43 | 0.02 | 0.02 | 0.07 | 0.53 | 0.21 | 0.78 | 15.79 |

## 2.2. Physical Performance Tests of the Palm Mat Geotextiles in the Lab

The weight, thickness, and tensile strength of palm mat geotextiles are important mechanical indexes for the laying application. Water permeability and saturated water content are beneficial for the infiltration and storage of rainwater, which affects the vegetation growth. Three pieces of palm mat geotextiles, each with an area of 1 m$^2$, were randomly selected from the product. The average

weight was measured by electronic balance (manufactured by the A&D company, Tokyo, Japan), having a precision of 0.01 g, and the average thickness was measured by an electronic Vernier caliber (manufactured by Mitutoyo company, Tokyo, Japan), having a precision of 0.01 mm. The tensile strength was measured by an electronic wire strength tester (manufactured by Sansi Electromechanical Company, Shanghai, China) and found to have a precision of 1 Pa. Sampling in the field was performed every three months with five replications per treatment. Three pieces of palm mat geotextiles, each with an area of 100 cm$^2$, were randomly selected from the product for observation and measurement of water permeability, saturated water content, and structural stability. The weight difference of the palm mat geotextiles before and after soaking in water (until the quality of geotextiles did not change) provided the saturated water content. Water permeability and structural stability were observed using simulated rainfall in the lab. The geotextile weight was measured every 30 min during simulated extreme rainfall, consisting of 30 mm/h for 3 h. (According to local meteorological data over the past three years, actual precipitation intensity was never more than 20 mm/h for 2 hours). Each experiment was repeated three times.

### 2.3. Field Experiments

### 2.3.1. Location of the Experimental Site

The experimental site is located in the north portion of the Hobq Desert, China, and the geographical coordinates are 108°42′28.13″ E, 40°29′49.70″ N (Figure 2). It is in a typical temperate continental arid monsoon climate with cold winters and hot summers. The annual average temperature is 6.7 °C, and the lowest and highest temperatures are –32.1 °C in January and 38.7 °C in July. Annual precipitation is 250 mm, with 75% of this total falling between July and September, and annual evaporation reaches 2400 mm. Average annual and maximum wind velocities are 3.5 and 16.6 m/s. The main wind directions are easterly (ENE and E, 116.22 VU) and westerly (SW, WSW, W, WNW, and NW, 104.71 VU) [11]. The field experiment began in June 2016 and lasted for one year.

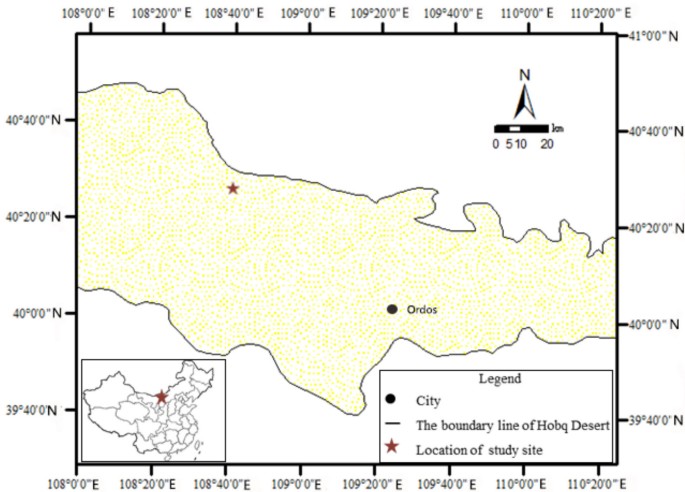

**Figure 2.** The experimental site is located in the north portion of the Hobq Desert, China, and the geographical coordinates are 108°42′28.13″ E, 40°29′49.70″ N.

### 2.3.2. Experimental Setting

We placed the palm mat geotextile group, the high-banded *Salix psammophila* sand barrier group, and the control group, with each having an area of 1500 m$^2$. We conducted a high-banded *S. psammophila* sand barrier experiment for comparison, because it is one of the more common and effective sand fixation measures in the Hobq Desert. Every group was repeated three times (Figure 3). The height and area of high-banded *S. psammophila* sand barriers was 20 cm and 1 × 1 m, respectively,

and the permeability was less than 5%. The quantity of seeds sown for the high-banded *S. psammophila* sand barrier group and for the control group was the same as that for the palm mat geotextile group.

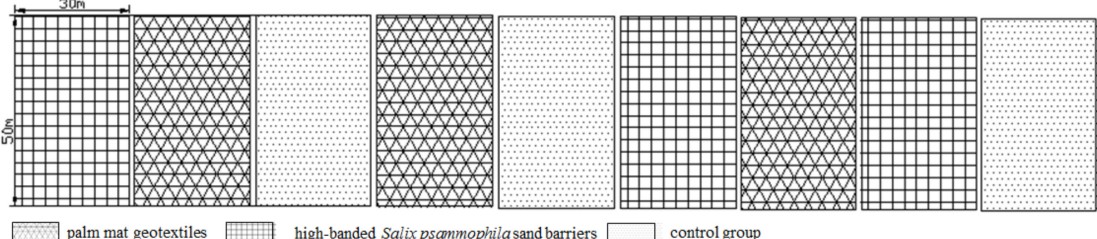

palm mat geotextiles    high-banded *Salix psammophila* sand barriers    control group

**Figure 3.** Schematic plan of test plots. We placed the palm mat geotextile group, the high-banded *S. psammophila* sand barrier group, and the control group, with each having an area of 1500 m$^2$ and the three replications per treatment.

Soil temperature and moisture directly affect the sprouting and growth of plants. Thus, for the upper 0–20 cm of the soil depth, temperature was continuously recorded in 5 cm steps with three replications; for the upper 0–30 cm of the soil depth, moisture was continuously recorded in 10 cm steps by a desert weather station, having a precision of 0.01 °C (manufactured by Sunshine Meteorological Science and Technology Company, Jingzhou, China), to record changes in soil temperature and moisture.

Three samples (1 × 1 m) were randomly selected and fixed from each group, and plant growth was continuously observed. The number of plants alive was recorded every month. Photographs of samples were taken by vertical photography every 3 months, and the same color vegetation pixels were automatically selected in Adobe Photoshop. Vegetation coverage is the proportion of vegetation pixels to the total pixels in the photograph. At the end of the experiment, all plants were dried in an oven for 48 hours at 80 °C and weighed to calculate the biomass.

### 2.4. Statistical Analysis

Tensile strength of the palm mat geotextiles, the loss rate of the substrate, the difference in soil temperature and moisture between different treatments in the same month, and indexes of the plant growth, including emergence ration, seeding number, vegetation coverage, and biomass, were evaluated based on statistical analysis of the data by ANOVA (PASW Statistics version 18.0, SPSS, Inc. 2009, Chicago, USA.). Assumptions of normality and homogeneity of variance were tested using the Shapiro–Wilk and Leneve tests, respectively. Since the data conform to these assumptions (Shapiro–Wilk, $P > 0.05$; Leneve tests, $P > 0.05$), Least significant difference and Student–Newman–Keuls tests were used to test for significant differences among treatments. Significance was accepted at the $P < 0.05$ level. All statistical analyses were performed in PASW Statistics version 18.0 (SPSS, Inc. 2009, Chicago, USA).

## 3. Results

### 3.1. Physical Performance of the Palm Mat Geotextiles

The average weight, thickness, and tensile strength of the palm mat geotextiles were 2023 ± 6.64 g/m$^2$, 20.14 ± 0.58 mm, and 860 ± 2.33 kPa, respectively. Tensile strength decreased by only 2% one year later. (Figure 4). During the experimental period, the palm mat geotextiles fixed sand dunes completely and without any breakage.

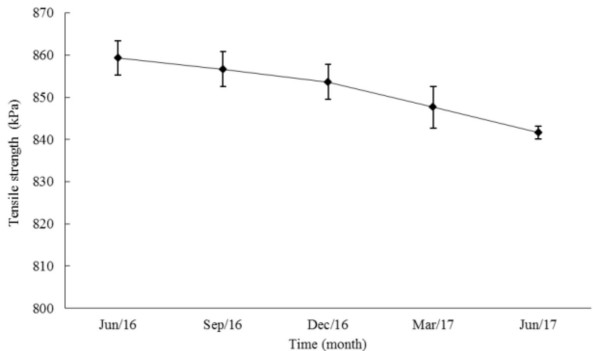

**Figure 4.** Tensile strength of palm mat geotextiles was measured by an electronic wire strength tester every 3 months from June 2016 to June 2017 in Hobq Desert Technology Research Institute of Inner Mongolia. Bars indicate standard errors of the mean (n = 5).

### 3.2. Water Permeability, Saturated Water Content, and Structural Stability

The results of the immersion experiments showed that the initial and postimmersion mass of the 100 cm$^2$ palm mat geotextiles were 20.11 ± 0.68 g and 65.78 ± 2.25 g and that the gravimetry of water content reached 227%. The results of the artificial simulated precipitation scour test showed that the substrate loss increased with the precipitation time in the first 1.5 h and the substrate loss was very small after 1.5 h. The total loss rate was 2.9%. (Figure 5).

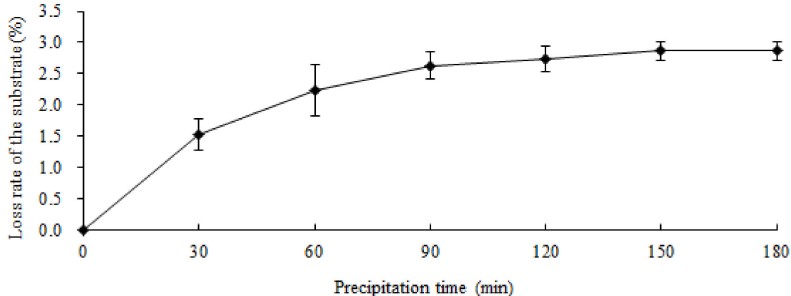

**Figure 5.** The loss proportion of the substrate. The geotextiles weight was measured every 30 min during simulated extreme rainfall, consisting of 30 mm/h for 3 h in Hobq Desert Technology Research Institute of Inner Mongolia. Bars indicate standard errors of the mean (n = 5).

### 3.3. Effects of the Palm Mat Geotextiles on Soil Temperature

The palm mat geotextiles had a great impact on soil temperature in the 0–5 cm depth range, with a smaller influence deeper in the soil. The soil temperature in the 0–5 cm depth range was significantly ($F_{(2,6)} = 5.14$, $P < 0.001$) lower than that in the high-banded *S. psammophila* sand barrier group and the control group in the summer, and significantly ($F_{(2,6)} = 5.14$, $P < 0.001$) higher than that in the control group in the winter. Little difference in the temperature curve existed between the high-banded *S. psammophila* sand barrier and control groups (Figure 6). Moreover, the soil temperature curve for the 0–5 cm depth range was essentially identical to that for the 5–10 cm, 10–15 cm, and 15–20 cm depth ranges. The high-banded *S. psammophila* sand barriers had little influence on the soil temperature.

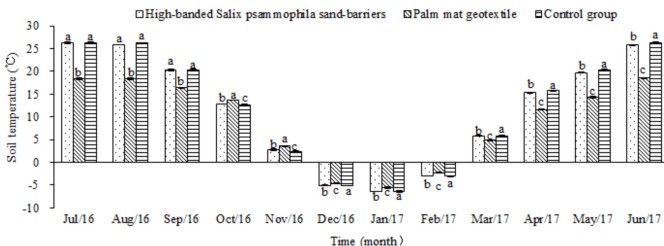

**Figure 6.** The soil temperature in 0–5 cm depth. The soil temperature of the treatments was continuously recorded in 5 cm steps for the upper 0–20 cm of the soil depth from June 2016 to June 2017. Bars indicate standard errors of the mean (n = 3). Note: Different lowercase letters represent significant difference between different treatments in the same month (P < 0.05).

*3.4. Effects of the Palm Mat Geotextiles on Soil Moisture*

The palm mat geotextiles had a great impact on soil moisture at a depth of 10 cm, with the influence being smaller with increasing depth. The soil moisture of the palm mat geotextiles at a depth of 10 cm was 2.9 times higher than that of the control group and 3.5 times higher than that of the high-banded *S. psammophila* sand barrier group in August ((F (2,6) = 5.14, P < 0.001)) (Figure 7). Therefore, the palm mat geotextiles can significantly improve the soil moisture. The soil moisture of the high-banded *S. psammophila* sand barrier group was slightly lower than that of the control group because increased numbers of plants in the high-banded *S. psammophila* sand barrier group caused the formation of more evaporation channels in the soil.

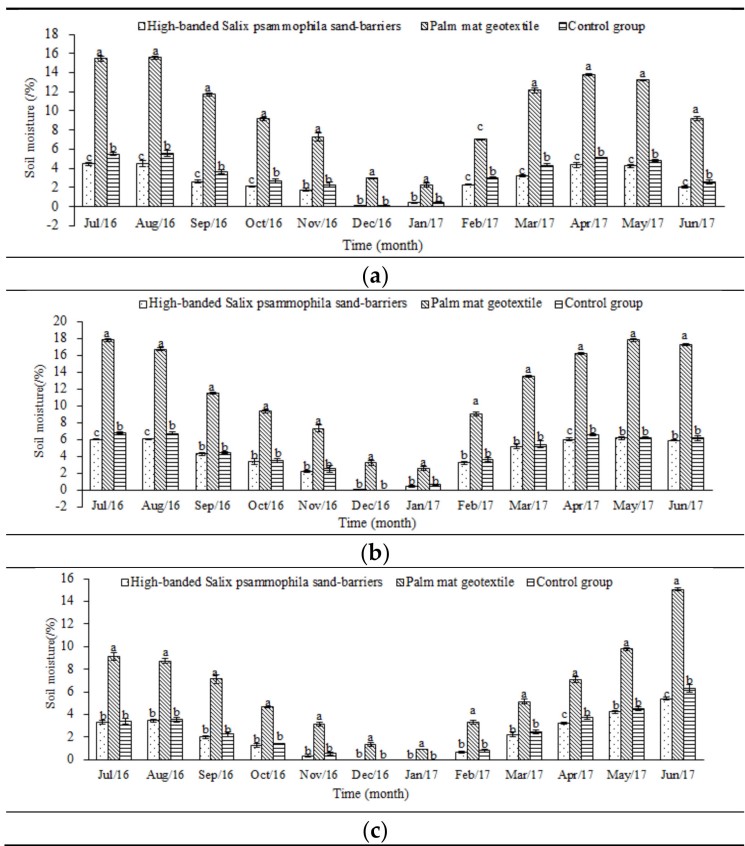

**Figure 7.** The soil moisture at (**a**) 10 cm, (**b**) 20 cm, and (**c**) 30 cm. The soil moisture of the treatments was continuously recorded in 10 cm steps for the upper 0–30 cm of the soil depth from June 2016 to June 2017. Bars indicate standard errors of the mean (n = 3). Note: Different lowercase letters represent significant difference between different treatments in the same month (P < 0.05).

### 3.5. Effects of the Palm Mat Geotextiles on Plant Growth

The effectiveness of the palm mat geotextiles largely depends on the vegetation restoration effect. The emergence ration among the palm mat geotextile group, the high-banded *S. psammophila* sand-barriers, and the control group were significantly different ((F (2,6) = 5.14, P = 0.001)) (Figure 8a). The palm mat geotextiles and high-banded *S. psammophila* sand barriers had a certain degree of wind erosion resistance, but the thickness of the palm mat geotextiles negatively influenced the emergence rate.

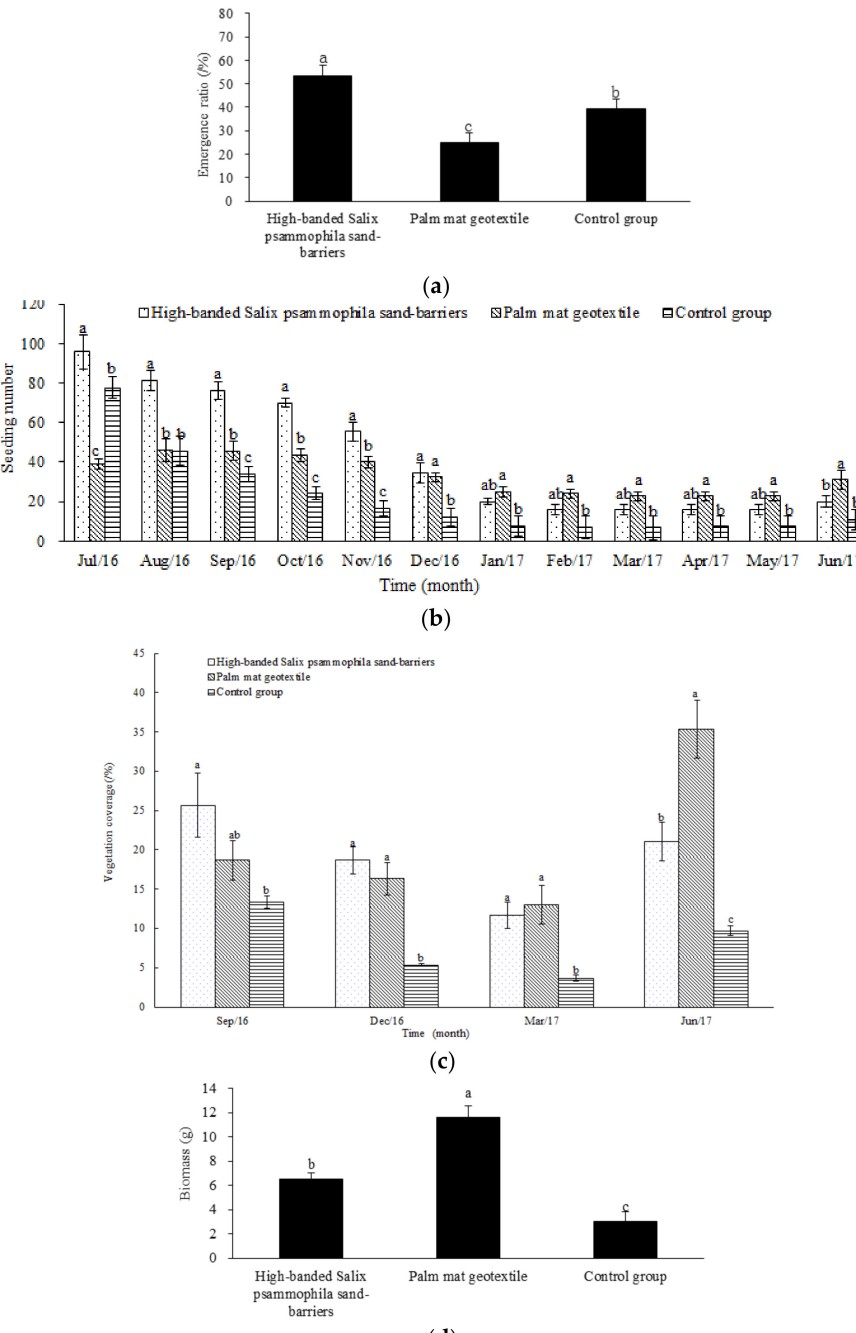

**Figure 8.** Index of the plants growth: (**a**) emergence ration, (**b**) seeding number, (**c**) vegetation coverage, and (**d**) biomass. Note: Different lowercase letters represent significant difference between different treatments in the same month (P < 0.05).

The seeding numbers among the palm mat geotextile group, the high-banded *S. psammophila* sand barrier group, and the control group were significantly different ((F (2,6) = 5.14, P < 0.001)) in July. Because of the high temperatures, drought, and wind erosion, the seeding numbers of the high-banded *S. psammophila* sand barrier group and of the control group sharply decreased, but the seeding numbers for the palm mat geotextile group increased from July to November. The seeding number for the three groups decreased to varying degrees from November to May of the following year. The number of seedlings of the three groups increased after May of the following year, and the number of seedlings for the palm mat geotextile group was 3.6 times higher than that of the control group and 1.5 times higher than that of the high-banded *S. psammophila* sand barrier group ((F (2,6) = 5.14, P = 0.018)) (Figure 8b). The vegetation coverage and biomass of the palm mat geotextile group were 3.9 and 4.1 times greater than that of the control group and were 1.7 and 1.8 times greater than that of the high-banded *S. psammophila* sand barrier group by the end of the experiment ((F (2,6) = 5.14, P < 0.001)) (Figure 8c,d). Moreover, the vegetation type of the palm mat geotextiles and high-banded *S. psammophila* sand barriers were mainly *A. desertorum* (Figure 9).

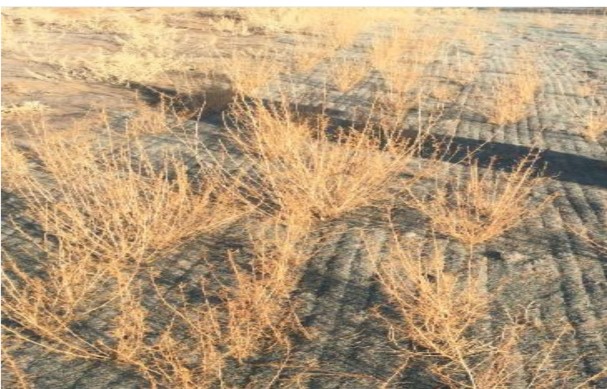

**Figure 9.** The vegetation type of the palm mat geotextiles and high-banded *S. psammophila* sand barriers were mainly *A. desertorum*.

The number of plants alive was recorded every month. Photographs of samples were taken by vertical photography every 3 months, and vegetation pixels were selected in Adobe Photoshop. Vegetation coverage is the proportion of vegetation pixels to the total pixels in the photograph. All plants were dried in an oven for 48 hours at 80 °C and weighed to calculate the biomass in June 2017. Bars indicate standard errors of the mean (n = 3).

## 4. Discussion

### 4.1. Physical Performance of the Palm Mat Geotextiles

In the course of construction and installation, geotextiles encounter various construction activities, such as stretching and fixing. These construction activities can lead to deformation and tension of geotextiles, which can improve the stability of slope fixation. Therefore, the tensile strength is of great significance to engineering applications [32]. Widely used materials for producing geotextiles include natural materials, represented by straw fibers, and synthetic materials, consisting of degradable chemical fibers of polyethylene and polypropylene [26]. The degradation products are oxalic acid and acetic acid, which are friendly to the environment [33]. The tensile strengths of straw mat geotextiles and coconut mat geotextiles measured in the laboratory were 104 and 134 kPa, respectively. After one year, the tensile strength of the two geotextiles decreased by 85 and 34%, respectively [34]. The three-dimensional geotextiles are made of polyethylene and polypropylene degradable materials, for which the maximum tensile strength does not exceed 320 kPa [35]. However, the tensile strength of the palm geotextiles used in this study was 860 kPa, which decreased by only 2% after one year. Straw checkerboard was weathered by rain and gradually became rotten, the general lifetime was

3 to 4 years [36]. Biological soil crusts is a new technology of sand fixation, but the crust is easily destroyed by external forces, and recovery takes about 5 years [37]. Chemical sand fixation technology is similar to biological soil crust technology, the crust is easily destroyed by external forces but cannot recover [38]. Therefore, the palm mat geotextiles are relatively stronger and more durable.

### 4.2. Water Permeability and Saturated Water Content of the Palm Mat Geotextiles

Desert areas are perennially drought prone and frequently lack water. Annual precipitation in the Hobq Desert is approximately 250 mm, while annual evaporation reaches 2400 mm. Even though there is occasional rainfall, most of the rainwater evaporates into the air because sand has little water storage capability and water retention ability. Therefore, most plants cannot survive in the desert. For the sake of plant survival, it is necessary to store enough water in the desert. Palm mat geotextiles have excellent water absorption and retention functions and slowly recharge water into ground near the surface by capillary action. The water content of the palm mat geotextiles reached 227% and provided a small "reservoir" for the plant. However, the shallow infiltration of rainwater leads to the decline of deep-rooted plants and the development of shallow-rooted semi-shrubs and herbs, which makes the natural system evolve into semi-artificial and semi-natural system [37].

### 4.3. The Effects of Palm Mat Geotextiles on Soil Temperature and Moisture

The surface soil moisture of the coconut and straw mat geotextile groups was always higher than that of the control group [34]. Our study also showed this. There are two main reasons why the soil temperature for the palm mat geotextile group was lower than that of the high-banded *S. psammophila* sand barrier group and control group in the summer: One is the shading effect of the geotextile group on the surface, providing protection from direct sunlight. The other is the higher soil moisture level in the geotextile area, which effectively reduced the soil temperature. In the winter, palm mat geotextiles covering the surface effectively reduced the surface wind speed and played a role in heat preservation. Two percent of the GS-3 sand-fixing agent measure had a cooling effect on soil temperature in the summer, the soil moisture of the chemical sand fixation t measure at a depth of 10 cm was 2 times higher than that of the control group in August in the Hobq Desert [39]. However, the soil moisture of the palm mat geotextiles at a depth of 10 cm was 2.9 times higher than that of the control group and 3.5 times higher than that of the high-banded S. *psammophila* sand barrier group in August

### 4.4. Effects of the Palm Mat Geotextiles on Plant Growth

2% GS-3 sand-fixing agent was used to stabilize sand dunes in the Hobq Desert, and, after one year, the vegetation coverage was 23% [39]. In another study, 1 × 1 m high-banded *S. psammophila* were used to stabilize sand dunes in the Hobq Desert, and, after one year, the vegetation coverage was 24% [40]. In our study, the vegetation coverage of the palm mat geotextiles was 35%, which indicated that palm mat geotextiles played a very important role in promoting plant growth. Traditional sand fixation measures use various materials, including plant straw, or chemical synthetic materials as grids for sand fixation, which cannot completely fix sand dunes and block near-surface sand flow; the palm mat geotextiles fully cover and completely stabilize sand dunes, effectively minimizing near-surface sand flow. In addition, the excellent water retention and heat preservation of palm mat geotextiles play a very important role in promoting plant growth.

### 4.5. The Cost and Application of Palm Mat Geotextiles

Palm is widely distributed in China, ranging from Eastern to Southwestern China, and it provides sufficient quantities of palm fiber. Currently, the price of palm mat geotextiles is approximately 3 CNY per square meter, including materials, transportation, and construction costs. High-banded *S. psammophila* sand barriers, which are widely used in the Hobq Desert, cost approximately 3.5 CNY per square meter. Sand-fixing agent measure costs approximately 5 CNY per square meter in the Hobq Desert [39]. Moreover, the sand fixing effect of palm mat geotextiles is superior to that of

high-banded *S. psammophila* sand barriers. With improvements in production technology, the price of palm mat geotextiles will continue to decline, which is favorable for increased usage and application over large areas.

Considering the harsh environments of desert areas and the characteristics of palm mat geotextiles, long-term field observations and research on the stability of its structure and performance, disintegration cycle, and impacts on plant growth are needed.

## 5. Conclusions

Physical performance tests generally showed that palm mat geotextiles have great flexibility and tensile strength and completely covered sand dunes for one year without any breakage. Immersion experiments showed that palm mat geotextiles had good water permeability, saturated water content, and structural stability after heavy rainfall. Our field experiments illustrated that the cooling effect in the summer, heat preservation in the winter, and the excellent resistance to wind erosion exhibited by palm mat geotextiles made it a good sand fixation material and one which is superior to high-banded *S. psammophila* sand barriers. Moreover, the cost of the palm mat geotextiles is cheaper than the high-banded *S. psammophila* sand barriers, which are widely used in the Hobq Desert. Therefore, palm mat geotextiles are a promising material for sand fixation in the Hobq Desert.

**Author Contributions:** Z.S. designed the experiments. Z.S., L.A., and D.H. conducted the experiments. Z.S. analyzed the experimental data and wrote the manuscript. H.Z. reviewed the manuscript. All authors have read and approved the manuscript.

**Funding:** This work was funded by the National Science and Technology Support Program (2015BAC06B01-01) and the National Natural Science Foundation of China (41371025)

**Conflicts of Interest:** The authors declare no conflict of interest. The funders had no role in the design of thestudy; in the collection, analyses, or interpretation of data; in the writing of the manuscript, or in the decision to publish the results.

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
