# Peer review of "Research on the Application of Palm Mat Geotextiles for Sand Fixation in the Hobq Desert"

_sustainability, doi:10.3390/su11061751_

Round 1

Reviewer 1 Report

After reading again the manuscript  "Sustainability - 436369 ", entitled    "Application of the palm mat geotextiles in the Hobq Desert,  N-China"   I am very satisfied seeing that it has been revised carefully. My critical question touching the economic part of the field experiment has been answered;  the other parts of the manuscript are - in my opinion - o.k.

Author Response

Dear professor,

I deeply appreciate for your careful review of my manuscript.Thank you very much for your approval of my manuscript.And thank you for your encouragement,I will continue my research in the future.

Yours sincerely,

Shuai Zhong

9th February 2019.

Reviewer 2 Report

The most important issues were solved by the authors. Please check the spelling and other formats.

Author Response

Dear professor,

I deeply appreciate for your careful review of my manuscript.The contents I revised carefully mainly include a thorough linguistic revision by MDPI(MDPI uses experienced, native English speaking editor,Full details of the edits  can be found in the manuscript) and refine the conclusion.At last, thank you very much for your guidance,I will continue my research in the furture.

Yours sincerely,

Shuai Zhong

9th February 2019.

Reviewer 3 Report

Please find my comments in the attached PDF file

Author Response

Dear professor,

I deeply appreciate for your careful review of my manuscript again, your valuable comments are very helpful to improve the quality of the manuscript. The contents I revised carefully mainly include a thorough linguistic revision by MDPI (MDPI uses experienced, native English speaking editor), deep discussion, adding the details such as statistical methods. All of the revisions can be found in the manuscript. At last, thank you very much for your guidance, I looking forward to your further valuable advice.

Yours sincerely,

Shuai Zhong

5th March 2019.

Reviewer 4 Report

I found the topic of this paper of interest. It provided a useful field trial of new product for sand stabilizing product. However, this paper needs a lot of editing by someone more proficient in English. Also, a lot of details are missing in the text, and captions for tables and figure need to be expanded. There is a lack of detail on statistical methods and its presentation in the results. 

See further comments made on the manuscript.

Author Response

Dear professor,

I deeply appreciate for your careful review of my manuscript.The contents I revised carefully mainly include a thorough linguistic revision by MDPI (MDPI uses experienced, native English speaking editor,full details of the edits  can be found in the manuscript),giving a cost comparison between different methods in the introduction, expanding all the captions for tables and figures,adding the details such as  statistical methods and vertical photography method, and refining the conclusion,all of the revisions can be found in the manuscript.At last, thank you very much for your guidance,I looking forward to your further valuable advice.

 Yours sincerely,

 Shuai Zhong

 9th February 2019.

Round 2

Reviewer 3 Report

Please check again your results for tensile strength. I am almost 100 % sure it should be "kPa" and not "Pa".

Reviewer 4 Report

I thank the authors for making the suggested edits and additions to your manuscript. It has greatly improved the quality of the manuscript. There are some additional minor questions/edits that require attention plus a couple of major issues that need to be addressed.

When I recommended that captions for figures and tables be expanded, I did not expect sections of the results to be cut and pasted. I have given examples of a suitable structure for captions and information captions should contain. Also, please look at some published papers for other examples of captions. I find it hard to believe this issue has not been brought to your attention before unless this is your first peer reviewed research paper?

More information is required on statistical methods (see comments in manuscript).

Author Response

Dear professor,

I deeply appreciate for your careful review of my manuscript again,your valuable comments are very helpful to improve the quality of the manuscript. The contents I revised carefully mainly include revising all the captions for tables and figures,adding the details such as statistical methods and presentation. All of the revisions can be found in the manuscript.At last, thank you very much for your guidance,I looking forward to your further valuable advice.

 Yours sincerely,

 Shuai Zhong

 25th February 2019.

Round 3

Reviewer 4 Report

I like to thank the authors for there attempt to satisfy the reviewers request for changes and additions to the manuscript which has improved the manuscript. However, some of the changes made were not correct or need editing plus there are some other issues. Forgive me for not mentioning earlier, but you should not discuss the results in the results section of the manuscript. (examples - line 245 "In general, the palm mat geotextiles had stability after heavy rains" and line 269 "Therefore, the palm mat geotextiles ...."

The references need to be checked to see if complete (example - line 603 What is the journal name?) 

I recommend the authors familiarise themselves with the format and style required for publishing a research paper in a peer-reviewed journal.

This manuscript is a resubmission of an earlier submission. The following is a list of the peer review reports and author responses from that submission.

Round 1

Reviewer 1 Report

Please find my comments in the attached PDF file.

Reviewer 2 Report

This paper deals with a major issue of the desertification process, which is the soil (sand) superficial loss and erosion. The idea is to test  a new material to prevent soil erosion without disturbing seed germination and plant emergence. This is a good set of data but there are some problems with the English style (needs a deep revision) and data presentation. Moreover, there is no statistical analysis to support the results and the discussion section is very poor.

Specific comments:

In Introduction (line 32) start the sentence with  “According to…” and not “As…”

In the abstract (line 18() include “respectively” after were,…

Remove et al. from the text

In line 56 change the text to: …. the lack of new technologies and mew materials.

Line 73: confusing. Rephrase

Materials and methods

The legends of tables 2 and 3 must be corrected and completed. In table 3, the authors provide some chemical parameters of the soil included in the raw material.

It is not entirely clear the comparison  of the new material and other materials (line 153-154). Did you check with the bibliography or did you perform some lab tests?

The legends of the figures are poor. For example in fig What is time/month? In several other figures change the label of the XX axis to month name, July, august etc

There are no statistics to compare treatments.

Discussion is poor. For example, the initial differences observed in several parameters at the beginning of the trial were not discussed. When did you start the experiment?

Authors contribution is missing…

Reviewer 3 Report

First few corrections,  second overall judgement:

L. 35            …  the world. Globally . . .

L. 36            …  km 2;  it covers…

L.38              … seriously desertification –affected . . .

L. 51/52       … difficult to cultivate it in the . . .   (better than “build”)

L. 52             … technique means to spray . . .

L. 60             …  used in landscaping, road and railway construction, reservoir and river embankment . .

L. 66              …  were used in . . .

L. 71              …  is located . . .      This is an important . . .

L. 75              …  sediment transported by floods . . .

L.78 /79        … sand fixation by palm mat geotextile, compared with  . . .

L.90; Tab. 2     Hedysarum mongolicum

L. 103            …  caliper ?  is it caliber ?

L. 111            …  during the simulation of extreme  . . .

L. 116            … coordinates are …

L. 120            …  falling between July and September

L. 174            … had a great  …

L. 178            … had a cooling . . .

L. 223/224    … Artemisia     (is double)

L. 250            …  to delete  “in the Hobq Desert, but the” . . .

L. 251            …The high-banded . . . . .  which are widely used …

L. 252            …  meter.  Moreover . . .

L. 265            … after heavy rainfall .

L. 284 /285   … to delete “and desertification”

The paper deals with a severe problem in semi-desert areas of the world known as desertification. The Hobq desert in Northern China represents a key area, and the authors have developed a very effective system to prohibit dune sand transport which threatens fields and infrastructure. The paper ist clearly structured; tables and fig’s are good, the same with the results which are broadly discussed. The list of references is (in my opinion) o.k.  Despite the overall good quality I have one critical question:

Traditionally, local people use since long high –banded salix psammophila sand barriers the construction of  them is relatively easy, and the raw material is locally available.      By contrast, palm mat geotextile is partly a high –tech product which is impossible to create without industrial support ( see good fig. 1). Therefore, production and transportation costs as well as such for raw material have to be included in the total calculation. Has this been done by the authors ?  They conclude that the price for one square m of the two completely different techniques is nearly the same , but with a clear advantage for the palm mat geotextile . This is difficult for the reviewer to understand ,and it would be helpful if the authors could give few additional information.